# Variability, timescales, and non-linearity in climate responses to black carbon emissions

Yang Yang[1*], Steven J. Smith[2*], Hailong Wang[1], Catrin M. Mills[1], Philip J. Rasch[1]

[1]Atmospheric Sciences and Global Change Division, Pacific Northwest National Laboratory, Richland, Washington, USA

[2]Joint Global Change Research Institute, Pacific Northwest National Laboratory, College Park, Maryland, USA

*Correspondence to yang.yang@pnnl.gov and ssmith@pnnl.gov

**Abstract**

Black carbon (BC) particles exert a potentially large warming influence on the Earth system. Reductions in BC emissions have attracted attention as a possible means to moderate near-term temperature changes. For the first time, we evaluate regional climate responses, non-linearity, and short-term transient responses to BC emission perturbations in the Arctic, mid-latitudes, and globally based on a comprehensive set of emission-driven experiments using the Community Earth System Model (CESM). Surface temperature responses to BC emissions are complex, with surface warming over land from mid-latitude BC perturbations partially offset by ocean cooling. Climate responses do not scale linearity with emissions. While stronger BC emission perturbations have a higher burden efficiency, their temperature sensitivity is lower. BC impacts temperature much faster than greenhouse gas forcing, with transient temperature responses in the Arctic and mid-latitudes approaching a quasi-equilibrium state with a timescale of 2–3 years. We find large variability in BC-induced climate changes due to background model noise. As a result, removing present-day BC emissions results in discernible surface temperature changes for only limited regions of the globe. In order to better understand the climatic impacts of BC emissions, both the drivers of non-linear responses and response variability need to be assessed across climate models.

## 1. Introduction

Black carbon (BC) aerosol, emitted from incomplete combustion, may be the second strongest positive anthropogenic climate forcing following carbon dioxide, which drew attention for potential climate mitigation from reducing BC emissions (Jacobson, 2004; Shindell et al., 2012; Bond et al., 2013; Smith and Mizrahi, 2013). The relationship between forcing and surface temperature changes caused by BC is complex and forcing is not a reliable indicator of the climatic impact of BC emissions (Stjern et al., 2017). BC absorbs solar radiation within the atmospheric column thereby warming the atmosphere with an influence on surface temperature that depends on its vertical location. At high altitudes, BC cools the surface by absorbing solar radiation (i.e., blocking it from reaching the surface) (Ramanathan and Carmichael, 2008), while BC at low altitudes warms the surface through diabatic heating (Ban-Weiss et al., 2012). In addition, heating the atmosphere and cooling the surface can increase atmospheric stability and therefore affect cloud formation, lifetime, and dynamical processes (Koren et al., 2004; McFarquhar and Wang, 2006; Koch and Del Genio, 2010). Through transformation from hydrophobic aggregates to hydrophilic particles coated with water-soluble substances (*i.e.,* aging processes), BC can become cloud-nucleating particles (Oshima et al., 2009), alter cloud microphysical processes, and suppress precipitation (Boucher et al., 2013). BC-induced warming or cooling can increase or decrease surface evaporation, resulting in further changes in precipitation and cloud formation (McFarquhar and Wang, 2006; Andrews et al., 2010; Ming et al., 2010; Ban-Weiss et al., 2012;

Kvalevåg et al., 2013). BC can also decrease surface albedo through deposition on
snow and ice, which is especially important to the climate at high latitudes and,
particularly the Arctic (Flanner et al., 2007; Qian et al., 2014) as snow/ice-albedo
effects are strong there. Taken together, these processes result in interactions
between BC and the atmosphere that can ultimately alter the net impact of BC on
climate, which have been termed rapid adjustments (Stjern et al., 2017).

Studies found that increases in BC emissions may contribute to the amplification

of Arctic warming directly by absorbing solar radiation in the atmosphere and
indirectly by reducing surface albedo through deposition on snow and ice (Flanner et
al., 2007; Qian et al., 2014). Flanner (2013) highlighted the importance of BC vertical
location in Arctic climate responses, with surface warming (cooling) due to BC in the
lower (upper) troposphere. In addition, BC outside the Arctic can influence the Arctic
climate through changing poleward heat transport. With BC snow/ice-albedo effect
excluded, Shindell and Faluvegi (2009) modeled an Arctic surface warming (cooling)
due to reducing (enhancing) mid-latitude BC atmospheric concentrations. Sand et al.
(2013a) found that this was due to the increased northward heat transport into the
Arctic. However, in another study where BC emissions were perturbed instead of
concentrations, Sand et al. (2013b) reported a decrease in northward heat transport
due to increases in mid-latitude BC emissions and suggested that the heating effect
of BC transported to the Arctic dominated the Arctic heating in the mid-latitude
perturbation simulation, leading to the opposite direction of atmospheric heat
transport compared to the concentration-driven perturbations. They also found that
increases in both BC emission and BC concentration in the Arctic atmosphere may
weaken poleward heat transport due to increasing Arctic temperature driven by BC
heating in the atmosphere and on snow and ice surfaces. Therefore, understanding
the Arctic climate impact of regional BC emissions is important for the Arctic climate
mitigation (Sand et al., 2016).

In order to archive a statistically significant signal for Arctic surface temperature

responses to BC emissions, Sand et al. (2013b) scaled present-day BC emissions
within the Arctic by a factor of 150 and emissions from mid-latitudes by a factor of 9,
respectively, in the NorESM model with BC snow/ice-albedo effects included. They
found that emissions of BC within the Arctic have an Arctic surface temperature
response 5 times larger than those from mid-latitudes and attributed this to BC
snow/ice-albedo feedbacks. The impact of BC emission perturbations on
mid-latitudes were not examined in that study, which we do in this work to contrast
the impact of BC on the Arctic with mid-latitudes.

Much of the previous work on BC has used atmosphere-only models or

prescribed BC concentrations (Hansen et al., 2005; Ming et al., 2010; Ban-Weiss et
al., 2012; Sand et al., 2013a), which artificially reduces variability in model results.
Results qualitatively differ between prescribed BC-concentration and emission-driven
simulations with coupled models (Sand et al., 2013a,b, 2015). A previous study using
coupled models found that the BC response in three of these models showed high
variability and inconsistency in the net sign of the responses to present-day BC
emissions both between models and even between ensemble members from the
same model (Baker et al., 2015). Stjern et al. (2017) investigated climate responses
to a tenfold increase in present-day anthropogenic BC concentrations or emissions
using five concentration-driven and four emission-driven global climate models. They
found that low-level cloud amounts increase, while higher-level clouds are diminished
for all models, which is dominated by rapid adjustments. The negative rapid
adjustments from changing clouds dampened positive instantaneous radiative forcing
of BC at the TOA, leading to a relatively small global surface warming. However, this
study did not consider response variability or non-linearity of responses. We note that
the model used in our study contains a different aerosol treatment (see below) than
the model used in Stjern et al. (2015).

To better understand the impacts of BC on climate, we present a comprehensive

analysis using a set of coupled simulations that examine regional climate responses,
non-linearity, and short-term transient climate responses to BC emission
perturbations. We focus in particular on the Arctic and also variability to assess if
climate responses to BC emission changes are likely to be discernable. Only
combustion and process-based anthropogenic BC emissions are perturbed, given
that the net global climate impact of open burning emissions has been assessed to be
small due to their high organic carbon fraction. A summary of key results is provided
below.

**2. Methods**
**2.1 Model description**
Here we use the fully coupled CESM (Community Earth System Model; Hurrell et
al., 2013) to simulate climate responses to BC emission perturbations. In
CAM5-MAM4 (Community Atmosphere Model version 5), the atmospheric
component of CESM, mass and number concentrations of aerosols are predicted
within four lognormal modes (i.e., Aitken, accumulation, coarse, and primary carbon
mode) of the modal aerosol module (MAM4; Liu et al., 2016). BC is emitted into the
primary-carbon mode and aged into the accumulation mode when coated with sulfate
or secondary organic aerosol. Particles in the accumulation mode, including BC and
other species, can serve as cloud condensation nuclei and have microphysical
effects on stratiform clouds and precipitation. The model physically treats
aerosol-cloud interactions using two-moment stratiform cloud microphysics, which
predicts number concentrations and mixing ratios of cloud water and ice (Morrison
and Gettelman, 2008; Gettelman et al., 2010). Activation of stratiform cloud droplets
is based on the scheme of Abdul-Razzak and Ghan (2000). In addition to the
standard treatments of aerosol-cloud interactions, we also include a set of
modifications that improves the simulation of aerosol wet scavenging and convective
transport (Wang et al., 2013). Although aerosols have no microphysical impact on
convective clouds, BC induced atmospheric heating can affect the ambient
temperature and convection. Convective precipitation can scavenge and remove
aerosols. Previous studies have extensively evaluated the CAM5 model simulations
of concentration, deposition, vertical profile and optical properties of BC (Wang et al.,
2013; Wang et al., 2015; Zhang et al., 2015a,b; Liu et al., 2016; Yang et al., 2017,
2018a,b) ), as well as climate variables (Hurrell et al., 2013; Yang et al., 2016a,b).
The model can simulate well the BC aerosol and climate variables in most regions of
the globe, but was reported to underestimate BC concentrations over China (Yang et
al., 2018a) and the Arctic (Wang et al., 2013) (although this earlier study used a
different emissions dataset), implying a possible underestimate of climate responses
to BC emissions in this study.
In our model simulations, atmospheric radiative transfer is calculated twice with
BC included and excluded, respectively. The changes in direct radiative effect and
cloud radiative effect induced by BC perturbation are calculated as $\Delta(F_{clear} - F_{clear,clean})$
and $\Delta(F_{clean} - F_{clear,clean})$, respectively, where $F_{clear}$ is the TOA flux calculated
neglecting scattering and absorption by clouds, $F_{clean}$ is the TOA flux calculated
neglecting scattering and absorption by BC, $F_{clear,clean}$ is the TOA flux calculated
neglecting scattering and absorption by both clouds and BC, and $\Delta$ refers to the
differences between the control and one of the emission perturbed simulations (Ghan,
2013). Note that these quantities include the impact of slow responses and feedbacks
(e.g., changes in sea surface temperature and sea ice and feedbacks with clouds) so
are not strictly comparable to the conventional definition of radiative forcing (Boucher
et al., 2013). The BC snow/ice-albedo effect on top of land and sea ice is included in
the model (Flanner et al., 2007; Yang et al., 2017, 2018c).
**2.2 Experimental configuration and emissions**
The following simulations are performed in this study. All insolation, greenhouse
gas concentrations and aerosol and precursor emissions, except BC, are fixed at
year 1850 levels, which include open burning emissions (van Marle et al., 2017).
The MID7X and ARC150X simulations use large emission perturbations to result
in signals large enough for detailed analysis. These regions are also particularly
important for BC impacts on the Arctic. The multipliers were selected following Sand
et al. (2013b) with the expectation that these would result in similar radiative
perturbations. This also allows a direct comparison to these previous results (Sand et
al., 2013b; and also Baker et al., 2015), which are also BC-emission simulations
using a coupled model with snow/ice-albedo feedbacks. The PD simulation then
allows us to evaluate the impact of present-day anthropogenic emissions. In brief, the
simulations conducted are:
1. PD: control simulation for BC in present-day conditions. BC emissions are

fixed at year 2010 (average of 2008–2012).

2. ARC150X: perturbed simulation to quantify the climate responses to Arctic BC

emissions. Same as PD except that year 2010 level anthropogenic BC

emissions over the Arctic (60–90°N) are scaled by a factor of 150.

3. MID7X: perturbed simulation to quantify the climate responses to mid-latitude

BC emissions. Same as PD except that year 2010 level anthropogenic BC

emissions over the mid-latitudes (28–60°N) are scaled by a factor of 7.

4.  ARC75X: perturbed simulation to quantify non-linearity of climate responses to Arctic BC emissions. Same as ARC150X except that Arctic BC emissions are scaled by a factor of 75.

5.  MID3.5X: perturbed simulation to quantify non-linearity of climate responses to mid-latitude BC emissions. Same as MID7X except that mid-latitude BC emissions are scaled by a factor of 3.5.

6.  MID14X: perturbed simulation to quantify non-linearity of climate responses to mid-latitudes BC emissions. Same as MID7X except that mid-latitude BC emissions are scaled by a factor of 14.

7.  PI: sensitivity simulation for BC in preindustrial conditions to compare results with Baker et al. (2015). BC emissions are at year 1850 levels.

Both mass and number of BC emissions are perturbed proportionally. Each simulation has one ensemble member for 100 years which are branched from year 81 of the PI simulation after 80 years spin-up, with the last 80 years used for most analysis. Another four short-term ensemble members for 30 years are conducted under both ARC150X and MID7X to examine the short-term transient climate response to BC emissions. These are branched from years 96, 112, 120, and 140 of PI simulation.

The CEDS (Community Emissions Data System) anthropogenic emissions (Hoesly et al., 2018) (version 2017-05-18) that were developed for the CMIP6 (Coupled Model Intercomparison Project Phase 6) model experiments are used in our simulations. Note that this emission dataset includes monthly BC emission

seasonality, which has been shown to be important for simulating BC in the Arctic
(Stohl et al., 2013). Figure S1 shows spatial distribution of annual anthropogenic BC
emissions for year 2010 (average of 2008–2012) and the regions for BC emission
perturbation. Over 60–90°N, anthropogenic BC emissions are mostly over the lower
latitude of the Arctic (60–70°N). Over the mid-latitudes, high BC emissions are mainly
located over eastern China. The annual total anthropogenic BC emission from the
Arctic in year 2010 is 0.08 Tg C yr$^{-1}$, with 70% contributed by energy sector. Scaled
by a factor of 150, ARC150X has 12.63 Tg C yr$^{-1}$ more BC emissions than the PD in
the Arctic. About 3.46 Tg C yr$^{-1}$ of BC is emitted from the mid-latitudes, with the
largest contribution from the residential sector (36%). With a scaling factor of 7,
MID7X includes an additional 20.74 Tg C yr$^{-1}$ of BC emission from mid-latitudes, as
compared to PD. Global annual anthropogenic BC for PD is 7.72 Tg C yr$^{-1}$, much
higher than 0.92 Tg C yr$^{-1}$ for PI.

**3. Regional climate responses to increases in Arctic and mid-latitude BC**
**emissions**
We first examine results from simulations with large perturbations of Arctic and
mid-latitude BC emissions (ARC150X and MID7X). Our initial simulations focused on
these regions due to the potentially high sensitivity of the Arctic to BC emissions.
Figure 1 presents the increases in annual, zonal-mean BC concentrations from
ARC150X and MID7X simulations, as compared to PD. Both Arctic and mid-latitude
BC emissions lead to BC concentration increases in the entire Northern Hemisphere,
with Arctic emissions mainly impacting low altitudes within the Arctic. In ARC150X,
due to extremely low temperatures at the surface and therefore temperature
inversions and a transport barrier (so called Arctic front), BC concentration increases
are mainly located over low altitudes within the Arctic. In MID7X, increased
mid-latitude emissions produce large increases in BC concentrations between 30°–
45°N. BC emitted over the mid-latitudes, which is lifted above the boundary layer and
transported at higher altitudes into the Arctic, leading to increased concentrations of
BC in the Arctic atmosphere. This spatial pattern is similar to that in Sand et al.
(2013b).

To explore the importance of emissions from these source regions to BC column

burdens, Table 1 summarizes BC burden efficiency, which is defined as the changes
in regional mean column burden of BC produced by per unit emission change,
calculated by differences between the perturbed and PD simulation. Over the Arctic,
increases in Arctic local BC emissions lead to an Arctic column burden efficiency of
0.425 ±0.024 mg m$^{-2}$ (Tg yr$^{-1}$)$^{-1}$. The burden efficiency of mid-latitude emissions over
the mid-latitudes is 0.191 ±0.004 mg m$^{-2}$ (Tg yr$^{-1}$)$^{-1}$, less than half of the efficiency of
Arctic emission on Arctic burden due to lower precipitation and frequent temperature
inversion in the Arctic compared to mid-latitudes. While the relative impact of
mid-latitude emissions on the Arctic burden efficiency (0.106 ±0.004 mg m$^{-2}$ (Tg yr$^{-1}$)$^{-1}$)
is smaller than either of the above efficiencies, the 28 times larger present-day total
emissions from mid-latitudes (3.70 Tg yr$^{-1}$) than the Arctic (0.13 Tg yr$^{-1}$) dominate
column burden contributions.
Table 1 also summarizes the changes in BC direct radiative effect, cloud radiative
effect, and snow/ice-albedo forcing induced by these large BC perturbations. Note
that these values include feedback effects from the coupled system, so are not
comparable to conventionally defined radiative forcing values. The albedo change
due to BC deposition on snow and ice is responsible for a significant increase in
Arctic surface forcing in both perturbations, with far smaller changes per unit emission
in mid-latitudes. Positive changes in direct radiative effect are offset by negative
changes in cloud radiative effect from increases in low cloud in the Arctic and
decreases in mid-level and high cloud over the mid-latitudes, similar to previous
results with a tenfold increase in present-day anthropogenic BC emissions (Stjern et
al., 2017).
Forcing efficiencies for direct radiative effect, cloud radiative effect and
snow/ice-albedo forcing (i.e., forcings produced by per unit emission change) are also
summarized in Table 1. Over the Arctic, local emissions from the Arctic have 2–4
times higher forcing efficiencies than emissions from the mid-latitudes, suggesting
higher impacts of a unit Arctic BC emission change to Arctic energy balance. Over the
mid-latitudes, although forcing efficiencies for direct radiative and cloud radiative
effects for Arctic emissions are 2–3 times lower than mid-latitude emissions, the
snow/ice-albedo forcing efficiencies are similar between Arctic and mid-latitude
emissions.
The annual mean surface air temperature response in ARC150X shows a
significant warming over both the Arctic and mid-latitudes (Figure 2). MID7X shows
temperature increases over the Arctic and most of the mid-latitude land regions, while
surface temperature decreases over some oceanic and coastal areas. The presence
of areas with both surface warming and cooling decreases the net average
temperature change over mid-latitudes. The seasonal mean surface air temperature
responses present similar spatial patterns (Figure S2), but slightly different
magnitudes (Figure S3). Over the Arctic, the warming due to Arctic BC emissions is
weakest in boreal summer. This is because the smaller summer sea ice and snow
fraction in the Arctic weakens BC snow/ice-albedo forcing. However, in the
mid-latitudes, warming is strongest in boreal summer for both Arctic and mid-latitude
BC emissions, because of stronger summer solar insolation and, therefore, stronger
BC heating in the atmosphere.
Due to the increased atmospheric absorption from BC, northward heat transport
for both perturbations decreases (Figure 3), consistent in sign with the results of Sand
et al. (2013b). The increases in temperature but decreases in net northward heat
transport indicate that the heating induced by changes in BC direct radiative effect
and BC snow/ice-albedo forcing dominate the overall BC-induced changes in energy
balance over the Arctic and mid-latitudes.
Arctic emissions are more efficient at impacting Arctic surface air temperatures
with an Arctic temperature sensitivity to Arctic emissions (0.169 ±0.052 K (Tg yr$^{-1}$)$^{-1}$)
seven times as large as the Arctic temperature sensitivity to mid-latitude emissions
(0.023 ±0.038 K (Tg yr$^{-1}$)$^{-1}$). Mid-latitude emissions, however, are likely to have a
larger present-day impact overall due to their 35 times larger preindustrial to
present-day emission increase (2.874 Tg yr$^{-1}$) than Arctic emissions (0.082 Tg yr$^{-1}$).
Note that, the Arctic temperature sensitivities are about 30% and 50% smaller than
those found in the coupled NorESM model experiments of Sand et al. (2013b) for
Arctic and mid-latitude emission perturbation simulations, respectively, probably due
to different model parameterizations and/or different vertical profile of BC driving the
net effect of BC impact on Arctic surface temperature (Flanner, 2013).

The vertical distribution of annual, zonal mean temperature responses (Figure 4)

shows that the ARC150X leads to a strong warming from the surface to 400 hPa over
the Arctic and between 40°–60°N. In MID7X, although the zonal mean surface
temperature response is relatively weak compared to ARC150X, a significant
warming is found in mid-latitudes between 500 and 200 hPa. BC transported from
mid-latitudes into the Arctic at high altitudes also results in Arctic temperature
increases aloft, between 400 and 300 hPa.

These changes in temperature pattern can change the stability of the atmosphere

and impact atmospheric circulation, as shown in Figure 5. Increases in BC emissions
over both the Arctic and mid-latitudes exert anomalous upward motions in the Arctic
and downward motions over the mid-latitudes, but for different reasons. In ARC150X,
stronger warming at the Arctic surface, compared to high altitudes, likely due to the
BC snow/ice-albedo effect, produces anomalous upward motions in the Arctic and
compensating downward motions between 50°–60°N. In MID7X, the stronger BC
warming at higher altitudes in mid-latitudes increases atmospheric stability and leads
to strong anomalous downward motions between 40°–60°N and compensating
upward motions over the Arctic and 10°–30°N (Johnson et al., 2004). Increasing
surface temperature and anomalous upward motion over the Arctic can weaken the
Arctic front, and the anomalous downward motion over the mid-latitudes favors air
stagnation.

Because of the anomalous downward motions over mid-latitudes in both

ARC150X and MID7X, high and/or mid-level cloud fraction decrease over
mid-latitudes (Figure 6). Due to slow feedbacks from increases in surface
temperature in the Arctic (Figure 2) and decreases in snow and sea ice, low cloud
fraction increases in the Arctic for both ARC150X and MID7X. The increases in low
cloud over mid-latitude oceans, which cause the cooling noted above, are due to
rapid adjustments as free-tropospheric BC heating reduces mixing with dry air above
the BC layer and increases the amount of marine stratocumulus (Johnson et al., 2004;
Sand et al., 2013a; Stjern et al., 2017).

Figure 7 shows changes in the total precipitation rate for the perturbed

simulations. Increases in Arctic and mid-latitude BC emissions lead to significant
decreases in precipitation over 60°N and 30–50°N, respectively, in correspondence
with anomalous downward motions (Figure 5) and decreases in mid-level and high
clouds (Figure 6) over these regions. Averaged over the Arctic and mid-latitudes,
changes in precipitation are weak, compared to uncertainties, except for the
mid-latitude precipitation response to BC emitted from mid-latitudes. The mid-latitude
precipitation sensitivity is $-7.67$ ($\pm3.34$) µm day$^{-1}$ (Tg yr$^{-1}$)$^{-1}$ for MID7X. Another
feature of the precipitation response is related to a northward shift in the ITCZ
(Intertropical Convergence Zone) in MID7X, which is consistent with the
hemispherically asymmetric warming pattern driven by increases in mid-latitude BC
emissions (Hwang et al., 2013; Baker et al., 2015).

Both ARC150X and MID7X show significant decreases by 13% and 3%,

respectively, in fractional area covered by sea ice over the Arctic, as compared to PD
(Figure S4). The snow depth over land also decreases, especially over Greenland.
The water equivalent snow depth averaged over Arctic land decreases by 5.0 cm (27%
relative to PD) and 0.8 cm (4%), respectively, for ARC150X and MID7X.

**4. Non-linearity of climate responses.**

We also evaluated the linearity of these responses by testing different emission

perturbation sizes. Figure 8 shows burden efficiencies, temperature and precipitation
sensitivities from simulations with Arctic BC emissions scaled by 75 and 150,
respectively, and mid-latitude BC emissions scaled by 3.5, 7 and 14, respectively,
with values summarized in Table 2. Stronger emission perturbations have a higher
burden efficiency. Over the Arctic, this is caused by anomalous Arctic upward
motions that weaken the Arctic front, lifting BC higher and leading to a longer BC
lifetime together with easier transport into the Arctic (Figure S5). Over mid-latitudes,
anomalous mid-latitude downward motions favor stagnation, which in turn
accumulates more BC in the atmosphere, together with decrease in precipitation (and
wet removal rate), contributing to increases in burden efficiency. All differences in
burden efficiencies between simulations with different emission perturbation sizes are
statistically significant with 95% confidence.

Despite this higher burden efficiency, the efficiency (per unit emission) of the

direct radiative effect decreases slightly. This is because strong BC perturbations
lead to more BC suspended in the atmosphere. More BC increases the attenuation of
the transmitted radiation, leading to a decrease in efficiency of BC light absorption in
the lower atmosphere and leading to a lower efficiency of direct radiative effect for a
stronger BC emissions perturbation.

The temperature sensitivity is lower, with 95% significance, for stronger

emission perturbations for both mid-latitude and Arctic BC between ARC75X and
ARC150X, as well between MID7X and MID14X (Table 2). The BC snow/ice-albedo
effect is found to be the most important factor in influencing Arctic temperature (Sand
et al., 2013b). Larger temperature increases from stronger BC emission perturbations
speed up sea ice and snow melt, leading to a weaker annual mean snow/ice-albedo
effect per unit BC emission for both Arctic and mid-latitudes. Therefore, the BC
snow/ice-albedo effect is more efficient for weaker emission perturbations, i.e., 0.151
($\pm$0.011) vs. 0.099 ($\pm$0.006) (W m$^{-2}$ (Tg yr$^{-1}$)$^{-1}$) for ARC75X and ARC150X and 0.026
($\pm$0.002) vs. 0.020 ($\pm$0.002) (W m$^{-2}$ (Tg yr$^{-1}$)$^{-1}$) for MID7X and MID14X of Arctic BC
snow/ice-albedo forcing efficiencies. All snow/ice-albedo forcing efficiency
differences are statistically significant. Together with lower efficiency of the direct
radiative effect, these explain the lower temperature sensitivity for stronger emission
perturbation. The non-linearity in snow-ice feedback relative to emissions size
appears to be the primary driver of surface temperature response non-linearity in
these results.

Additional evidence for BC non-linearity can be found in the literature. Sand et al.

(2015) simulated climate responses to BC in NorESM with present-day emissions
multiplied by 25 and reported that the changes in TOA net shortwave flux was 7.5
(±0.3) W m$^{-2}$ relative to preindustrial conditions and the temperature response was
1.2 (±0.1) K. If we assume a linear emission-response relationship, present-day BC
would cause an inferred shortwave flux and surface temperature change of 0.312
(±0.013) W m$^{-2}$ and 0.050 (±0.004) K in Sand et al. (2015), much lower than the 0.552
W m$^{-2}$ and 0.141 K found in Baker et al. (2015) for a present-day BC emission
perturbation with essentially the same model. Note that the change in shortwave flux
is not proportional to the surface temperate change, further emphasizing that forcing
is not a good predictor of surface temperature change for BC. This comparison is
consistent with our finding that temperature sensitivity is lower for stronger BC
emission perturbations. We note, however, that emission datasets with different
spatial distributions and seasonality were used in those two experiments (because of
this the difference in global emissions between the two experiments is about 17, not
25 times). While this might impact the magnitude of model responses, it is unlikely to
change the overall conclusion of a substantially different temperature response to
current-day emissions as compared to a 17–25 times larger BC perturbation.

The mid-latitude shows significantly stronger precipitation sensitivity for a

stronger perturbation, comparing MID7X and MID14X, which is consistent with the
higher burden efficiency. This is in the opposite direction to the surface temperature
sensitivity. Variability in MID3.5X is larger than the mean value for both temperature
and precipitation sensitivity, which highlights the challenge of testing differences for
smaller BC perturbation magnitudes. Note that the impact of BC on clouds and
precipitation is uncertain, especially in the Arctic, due to the limited treatment of Arctic
clouds in climate models (McFarquhar et al., 2011). These results suggest that in
order to examine the climate responses to BC emissions in short-term climate model
simulations, a large emission perturbation is needed to get a clear signal, but
non-linearity of the responses also needs to be evaluated.

**5. Short-term transient climate responses**

To assess the short-term transient climate responses to BC emissions, Figure 9

shows surface temperature responses to BC emissions from ARC150X and MID7X
for the first 30 years averaged over five short ensemble members. We also show a
numerical fit to the short-term transient response using a Hamiltonian Monte Carlo
technique (Betancourt, 2017). We fit to the following form:
$$T_{ave}\,(1 - e^{-t/\tau}),$$
where we have constrained the fit to converge to the long-term average temperature
response ($T_{ave}$) by our finding that there is no detectable long-term trend after the
initial transient period over a 100-year time horizon.

Over both the Arctic and mid-latitudes, transient temperature responses quickly

approach a quasi-equilibrium state. Transient timescales ($\tau$) for the ARC150X
perturbation were estimated to be 2.7 (2.0, 3.4) years, while the mid-latitude
timescales for the ARC150X and MID7X perturbations were 1.8 (1.1, 2.2) and 2.9
(1.2, 4.2) years respectively (brackets provide 10-90% fitting intervals). The Arctic
response to MID7X was too noisy to produce a fit. These timescales are shorter than
those in a global BC perturbation experiment (Sand et al., 2015), which is expected
as ocean thermal inertia would play a larger role globally as compared to the Northern
Hemisphere or Arctic. The BC response timescales here are also shorter than those
seen from $CO_2$ concentration steps in GCMs (Geoffroy et al., 2013). There is also no
long-term temperature increase, at least over a 100-year time horizon, after the initial
transient period. A linear fit over years 10-100 for the perturbation responses results
in no statistically significant linear trends for any of the four perturbations (see SI
code).

Note that the average of even five ensemble members shows oscillatory behavior

due to the imposition of a step BC emission perturbation. This oscillatory behavior
degrades our ability to quantify the perturbation response timescale. In future work, a
linearly phased-in perturbation might result in a cleaner signal for determining the
initial response time-scale.

**6. Climate responses to present-day anthropogenic BC emissions**

Baker et al. (2015) showed that the climate responses to BC emissions had very

large uncertainties based on results from four global models. Here, we also quantified
the impact of present-day anthropogenic BC emissions (Figure 10) by comparing a
present-day (PD) and pre-industrial (PI) simulation conducted with the
CESM-CAM5-MAM4 model used in this work. PD emissions produce statistically
significant surface air temperature changes over only limited regions in the Northern
Hemisphere. Decreased temperatures are found over eastern China, South Asia,
North Atlantic Ocean, and North American Arctic, partly due to cloud changes driven
by BC rapid adjustments. Increased temperatures are found over the Tibetan Plateau,
Greenland and high-latitude land regions likely because of the BC snow/ice-albedo
effect (Figure S6).

The spatial pattern is similar to that from the ECHAM6-HAM2 in Baker et al.

(2015). Although CESM-CAM5-MAM4 also includes the BC snow/ice-albedo effect,
we do not see the strong warming produced in NorESM under present-day BC
emissions. In Baker et al. (2015), NorESM had a global net TOA shortwave forcing
efficiency of 0.076 W m$^{-2}$ (Tg yr$^{-1}$)$^{-1}$, nominally higher than 0.043 ±0.073 W m$^{-2}$ (Tg
yr$^{-1}$)$^{-1}$ calculated in this study with CESM-CAM5-MAM4, although the difference is
well within one standard deviation. Longer model runs would be needed to determine
if the BC snow/ice-albedo effect is significantly different in CESM and NorESM. In
addition, there may be a small contribution from a shorter BC lifetime (7.22 days in
CESM-CAM5-MAM4 Vs. 7.82 days in NorESM) that might also help explain the
weaker warming in CESM-CAM5-MAM4 as compared to NorESM.

We find that variability is substantial in our experiments. Although statistically

significant surface temperature changes are found regionally, as mentioned above,
large-scale global surface temperature change from current-day BC emissions is
statistically indistinguishable from zero (0.006 ±0.238 K globally and 0.020 ±0.346 K
for land only). The global temperature response is within the range of –0.085 to 0.152
K from the four models in Baker et al. (2015). Even in the large MID7X perturbation,
variability is still fairly large relative to the signal (0.45 ±0.27 K for mid-latitude
temperate change) and would overwhelm any large-scale signal for more realistic
perturbation sizes. Similarly, while the mid-latitude precipitation response to
mid-latitude BC emissions is strong for a MID7X perturbation, this would be difficult to
detect for a present-day perturbation.

**7. Conclusions and discussions**

BC has been estimated to potentially have one of the largest positive (warming)

anthropogenic forcing influences. As a result, there has been substantial scientific
and policy attention focused on the potential for BC to moderate climate change in the
near-term. In this study, for the first time, we conduct a comprehensive set of
emission-driven experiments using a leading coupled climate model (CESM). With a
comprehensive set of experiments, we examined regional climate responses,
non-linearity, and short-term transient responses to BC emission perturbations in the
Arctic, mid-latitudes, and globally.

With increases in mid-latitude BC emissions, surface air temperature increases

over land, while it decreases over oceanic and coastal areas. Increases in Arctic BC
emissions lead to warming over both the Arctic and mid-latitudes. Increases in Arctic
and mid-latitude BC emissions also decrease precipitation over 60°N and 30–50°N,
respectively. Arctic emissions are more efficient in influencing Arctic surface air
temperatures compared to mid-latitude emissions, with an Arctic temperature
sensitivity to Arctic emissions seven times as large as that to mid-latitude emissions.

We find that climate responses do not scale linearity with emissions. While

stronger BC emission perturbations have a higher burden efficiency, efficiencies of
snow/ice-albedo forcing and direct radiative effect are lower, leading to a lower
temperature sensitivity for stronger BC emission perturbations.

BC also impacts temperature much faster than greenhouse gas forcing, with

transient temperature responses in the Arctic and mid-latitudes approaching a
quasi-equilibrium state with a timescale of 2–3 years. While it has previously been
found that, globally, aerosols have a faster impact on temperature as compared to
greenhouse gases (Shindell., 2014), termed a "geometric effect" (Meinshausen et al.,
2011), we find here that BC perturbations have a very short response time,
particularly for Arctic and mid-latitude perturbations. This means that previous studies
which have implicitly assumed that the temperature response timescales of BC and
GHGs are the same (Boucher and Reddy 2008, Stohl et al., 2015) have likely
underestimated the short-term impact of BC emission changes.

We find large variability in BC-induced climate changes. Baker et al. (2015)

provided error bars of global temperature response for different models in their Figure
4a. We note, however, that the error bars in Baker et al. are underestimated because
of their assumption of independence of all annual data points (note their use of
$1\sigma/sqrt(N)$ in their error bars for Figure 4). Climate model surface temperatures are
strongly correlated over short time scales, which means that instead of the number of
data points, a more appropriate measure is the effective number of independent data
points ($N_{eff}$). The 100-year model runs examined here do not provide enough data for
this calculation. (Note that we were able to estimate $N_{eff}$ for the 300-year CESM
control run from CMIP5, which indicates that runs around 3 times as long as those
presented here may be necessary). We, therefore, present standard deviation as a
metric of variance.

The standard deviation (SD) of global mean surface temperature in PI, PD, and in

all of our perturbed simulations is around 0.17–0.19 K, indicating that the dominant
source of temperature variability is probably due to internal climate variability or
model noise. The SD for temperature responses in perturbed simulations relative to
PD are in ranges of 0.24–0.26, roughly 1.4 times the control run temperature
variability. This is in the expectation from subtracting two independent Gaussian
noise distributions. While there could be an additional contribution to variability from
BC-climate interactions, this appears to be small in this case given the relatively small
surface temperature response to BC. We also observe large variability in cloud
radiative effects, which we note may be impacted by interactions with BC.

While we have demonstrated non-linear responses at high emission levels, this

non-linearity is not sufficient to produce statistically significant global temperature
changes from present-day BC emissions in this model. Such non-linearities mean
that the implications of large BC emission perturbation experiments, such as recent
tenfold BC experiments (Stjern et al., 2017) for present-day conditions are unclear.
While snow/ice-albedo feedbacks appear to dominate the non-linear relationship in
these results, this may not be the case in other models.

Our results point to the importance of better quantifying the variability in BC

responses in the Earth system. We note that in the one model with a consistent PD-PI
signal in a set of recent PD-PI BC experiments (NorESM1-M), the size of that signal
is still smaller than the variability found here, based on similar SD in Arctic
temperature change for the ARC150X simulations in the two models (compare Figure
S3 here with Figure 9 in Sand et al. (2013b)). If the global variability of BC response
in NorESM is also similar to that in CESM, then the global average temperature
change from NorESM (0.141 K) (Baker et al., 2015) is also much smaller than the SD
in CESM of 0.238 K. However, we do note that there was a fair degree of consistency
in temperature change signal in NorESM between two ensemble members (0.129 K
and 0.152 K). This may mean that variability in the global BC response in the
NorESM model could be smaller than seen in our results due to the stronger NorESM
BC temperature response. Longer simulations would likely be required to assess this.
While, compared to temperature or precipitation, aerosol burdens, BC direct radiative
effects and snow/ice-albedo forcings have much larger signal to noise ratios, i.e. ratio
of mean response to standard deviation (Table 1), and can be useful as diagnostics,
BC forcing does not provide a reliable indicator of surface temperature changes
across models.

These results indicate that even substantial BC emissions reductions from

current levels may lead to detectable surface temperature changes for only limited
regions of the globe. Our results have significant implications for near-term climate
mitigation associated with BC as well as global and regional climate attribution. We
note that regional climate sensitivities (RCS), used as an approximate approach to
represent the impact of BC (Collins et al., 2013; Sand et al., 2016), are generally
evaluated using model simulations with prescribed forcing or burdens (Shindell and
Faluvegi, 2009), which artificially reduce response variability and imply a certainty in
BC responses that may not exist in reality. Variability within any given model run,
which has generally not been reported in current literature, is large relative to BC
responses. It is, therefore, not clear if current BC emission levels result in statistically
significant large-scale climatic changes. We suggest that impacts of BC on climate
should be expressed directly in terms of impacts per unit emissions (e.g. Table 1),
and not only relative to forcing given the complex relationship between BC climatic
impacts and top of the atmosphere forcing. In addition, BC impacts should be
re-evaluated using coupled models, and provided with measures of response
variability, such as standard deviation. In order to better assess the potential impact
of changes in BC emissions it is critical to quantify the non-linearity of BC response
efficiencies with respect to emission perturbation size in other models, as well as the
causes of those non-linearities.

***Data availability.***
All the emissions data sets used in this study can be obtained from
https://esgf-node.llnl.gov/search/input4mips/ (Hoesly et al., 2018; van Marle et al.,
2017). The CESM model is publicly available at
http://www.cesm.ucar.edu/models/cesm1.2/ (Hurrell et al., 2013). Our model results
can be made available through the National Energy Research Scientific Computing
Center (NERSC) servers upon request.
***Competing interests.***
The authors declare that they have no conflict of interest.
***Author contribution***.
YY, SJS and HW designed the research; YY performed the model simulations; YY,
and SJS analyzed the data. All the authors discussed the results and wrote the paper.
***Acknowledgments***.
This research is based on work that was supported by the U.S. Environmental
Protection Agency, and the U.S. Department of Energy (DOE), Office of Science,
Biological and Environmental Research as part of the Regional and Global Climate
Modeling program. The Pacific Northwest National Laboratory is operated for DOE by
Battelle Memorial Institute under contract DE-AC05-76RLO1830. The National
Energy Research Scientific Computing Center (NERSC) provided computational
support.

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

**Table 1.** Changes in black carbon (BC) column burden, direct radiative effect (DRE)
and cloud radiative effect (CRE) at the top of the atmosphere (TOA), surface BC
snow/ice-albedo forcing, surface temperature (T) and total precipitation rate (P,
including rain and snow) averaged over the Arctic (60–90°N), mid-latitudes (28–60°N)
and the globe between perturbed (ARC150X/MID7X) and PD simulations. BC burden,
DRE, CRE, and snow/ice-albedo forcing efficiencies, T sensitivity and P sensitivity
are calculated as changes in regional mean BC column burden, DRE, CRE,
snow/ice-albedo forcing, T and P divided by changes in global total BC emissions
between perturbed and PD simulations, respectively. 1-σ for 80-annual means is
shown in the parentheses. Note that these quantities include the impact of slow
responses and feedbacks (e.g., changes in sea surface temperature and sea ice and
feedbacks with clouds) so are not strictly comparable to the conventional definition of
radiative forcing.

| | ΔColumn Burden (mg m$^{-2}$) | | | Burden Eff. (mg m$^{-2}$ (Tg yr$^{-1}$)$^{-1}$) | | | ΔDRE (W m$^{-2}$) | | |
|---|---|---|---|---|---|---|---|---|---|
| | 60–90°N | 28–60°N | Global | 60–90°N | 28–60°N | Global | 60–90°N | 28–60°N | Global |
| ARC150X | 5.37 (±0.30) | 1.34 (±0.05) | 0.63 (±0.03) | 0.425 (±0.024) | 0.106 (±0.004) | 0.050 (±0.002) | 3.94 (±0.39) | 0.83 (±0.04) | 0.45 (±0.03) |
| MID7X | 2.19 (±0.09) | 3.97 (±0.09) | 1.26 (±0.03) | 0.106 (±0.004) | 0.191 (±0.004) | 0.061 (±0.001) | 2.90 (±0.19) | 2.49 (±0.09) | 1.00 (±0.04) |

| | DRE Eff. (W m$^{-2}$ (Tg yr$^{-1}$)$^{-1}$) | | | ΔCRE (W m$^{-2}$) | | | CRE Eff. (W m$^{-2}$ (Tg yr$^{-1}$)$^{-1}$) | | |
|---|---|---|---|---|---|---|---|---|---|
| | 60–90°N | 28–60°N | Global | 60–90°N | 28–60°N | Global | 60–90°N | 28–60°N | Global |
| ARC150X | 0.39 (±0.03) | 0.11 (±0.00) | 0.05 (±0.00) | −3.83 (±0.98) | −0.46 (±0.84) | −0.22 (±0.54) | −0.30 (±0.08) | −0.04 (±0.07) | −0.02 (±0.04) |
| MID7X | 0.17 (±0.01) | 0.22 (±0.01) | 0.08 (±0.00) | −2.30 (±0.96) | −3.16 (±0.90) | −1.26 (±0.51) | −0.11 (±0.05) | −0.15 (±0.04) | −0.06 (±0.02) |

| | ΔSnow/ice-albedo Forcing (W m$^{-2}$) | | | Snow/ice-albedo Eff. (W m$^{-2}$ (Tg yr$^{-1}$)$^{-1}$) | | | ΔT (K) | | |
|---|---|---|---|---|---|---|---|---|---|
| | 60–90°N | 28–60°N | Global | 60–90°N | 28–60°N | Global | 60–90°N | 28–60°N | Global |
| ARC150X | 1.26 (±0.08) | 0.12 (±0.02) | 0.10 (±0.01) | 0.099 (±0.006) | 0.010 (±0.002) | 0.008 (±0.001) | 2.13 (±0.65) | 0.78 (±0.22) | 0.48 (±0.26) |
| MID7X | 0.53 (±0.05) | 0.18 (±0.03) | 0.07 (±0.01) | 0.026 (±0.002) | 0.009 (±0.001) | 0.003 (±0.000) | 0.48 (±0.79) | 0.45 (±0.27) | 0.23 (±0.26) |

| | T Sensitivity (K (Tg yr$^{-1}$)$^{-1}$) | | | ΔP (mm day$^{-1}$) | | | P Sensitivity (μm day$^{-1}$ (Tg yr$^{-1}$)$^{-1}$) | | |
|---|---|---|---|---|---|---|---|---|---|
| | 60–90°N | 28–60°N | Global | 60–90°N | 28–60°N | Global | 60–90°N | 28–60°N | Global |
| ARC150X | 0.169 (±0.052) | 0.062 (±0.018) | 0.038 (±0.020) | −0.043 (±0.079) | −0.011 (±0.066) | 0.010 (±0.023) | −3.38 (±6.29) | −0.86 (±5.26) | 0.77 (±1.84) |
| MID7X | 0.023 (±0.038) | 0.022 (±0.013) | 0.011 (±0.012) | 0.048 (±0.096) | −0.159 (±0.069) | −0.032 (±0.022) | 2.34 (±4.61) | −7.67 (±3.34) | −1.52 (±1.04) |


**Table 2.** BC burden, DRE, CRE, and snow/ice-albedo forcing efficiencies, T
sensitivity and P sensitivity over the Arctic (60–90°N), mid-latitudes (28–60°N) and
the globe between perturbed (ARC75X/ARC150X/MID3.5X/MID7X/MID14X) and PD
simulations. 1-σ for 80-annual means is shown in the parentheses. Bold values
between two simulations (ARC75X/ARC150X, MID3.5X/MID7X, and MID7X/MID14X)
indicates statistically significant changes with 95% confidence from a two-tailed
Student's t test.

| | | ARC75X | ARC150X | | | MID3P5X | MID7X | MID14X |
|---|---|---|---|---|---|---|---|---|
| | | \multicolumn{7}{c}{Burden Eff. (mg m$^{-2}$ (Tg yr$^{-1}$)$^{-1}$)} | | | | | | |
| 60–90°N | | **0.406 (±0.021)** | **0.425 (±0.024)** | | | **0.095 (±0.005)** | **0.106 (±0.004)** | **0.124 (±0.004)** |
| 28–60°N | | **0.097 (±0.004)** | **0.106 (±0.004)** | | | **0.175 (±0.005)** | **0.191 (±0.004)** | **0.219 (±0.005)** |
| Global | | **0.047 (±0.002)** | **0.050 (±0.002)** | | | **0.055 (±0.001)** | **0.061 (±0.001)** | **0.070 (±0.002)** |
| | | \multicolumn{7}{c}{DRE Eff. (W m$^{-2}$ (Tg yr$^{-1}$)$^{-1}$)} | | | | | | |
| 60–90°N | | **0.346 (±0.036)** | **0.312 (±0.031)** | | | **0.146 (±0.014)** | **0.140 (±0.009)** | **0.137 (±0.006)** |
| 28–60°N | | **0.069 (±0.005)** | **0.066 (±0.003)** | | | **0.129 (±0.006)** | **0.120 (±0.004)** | **0.112 (±0.003)** |
| Global | | **0.038 (±0.003)** | **0.035 (±0.003)** | | | **0.051 (±0.003)** | **0.048 (±0.002)** | **0.046 (±0.001)** |
| | | \multicolumn{7}{c}{CRE Eff. (W m$^{-2}$ (Tg yr$^{-1}$)$^{-1}$)} | | | | | | |
| 60–90°N | | **−0.533 (±0.232)** | **−0.303 (±0.078)** | | | −0.091 (±0.166) | **−0.111 (±0.046)** | **−0.015 (±0.029)** |
| 28–60°N | | 0.010 (±0.222) | −0.037 (±0.067) | | | **0.070 (±0.203)** | **−0.152 (±0.043)** | **0.129 (±0.035)** |
| Global | | **−0.028 (±0.071)** | **−0.017 (±0.043)** | | | 0.013 (±0.058) | **−0.061 (±0.025)** | **0.035 (±0.010)** |
| | | \multicolumn{7}{c}{Snow/ice-albedo Eff. (W m$^{-2}$ (Tg yr$^{-1}$)$^{-1}$)} | | | | | | |
| 60–90°N | | **0.151 (±0.011)** | **0.099 (±0.006)** | | | **0.030 (±0.003)** | **0.026 (±0.002)** | **0.020 (±0.002)** |
| 28–60°N | | **0.013 (±0.003)** | **0.010 (±0.002)** | | | **0.011 (±0.002)** | **0.009 (±0.001)** | **0.007 (±0.001)** |
| Global | | **0.012 (±0.001)** | **0.008 (±0.001)** | | | **0.004 (±0.001)** | **0.003 (±0.000)** | **0.003 (±0.000)** |
| | | \multicolumn{7}{c}{T Sensitivity (K (Tg yr$^{-1}$)$^{-1}$)} | | | | | | |
| 60–90°N | | **0.239 (±0.116)** | **0.169 (±0.052)** | | | 0.042 (±0.098) | **0.023 (±0.038)** | **0.008 (±0.015)** |
| 28–60°N | | 0.067 (±0.032) | 0.062 (±0.018) | | | 0.020 (±0.025) | **0.022 (±0.013)** | **0.015 (±0.005)** |
| Global | | **0.040 (±0.035)** | **0.038 (±0.020)** | | | 0.008 (±0.033) | **0.011 (±0.012)** | **0.005 (±0.005)** |
| | | \multicolumn{7}{c}{P Sensitivity (μm day-1 (Tg yr$^{-1}$)$^{-1}$)} | | | | | | |
| 60–90°N | | −2.88 (±13.39) | −3.38 (±6.29) | | | 1.73 (±10.85) | 2.34 (±4.61) | 1.86 (±2.06) |
| 28–60°N | | −0.96 (±9.45) | −0.86 (±5.26) | | | −7.69 (±8.90) | **−7.67 (±3.34)** | **−8.53 (±1.61)** |
| Global | | 0.31 (±3.10) | 0.77 (±1.84) | | | −1.99 (±2.81) | −1.52 (±1.04) | −2.15 (±0.49) |


**Figures for Paper**

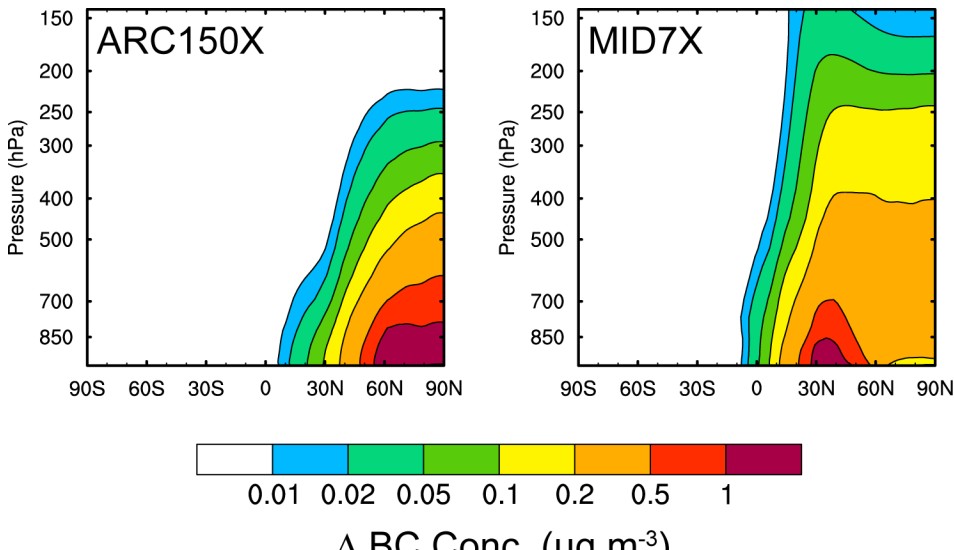



**Figure 1.** Difference in annual and zonal mean BC concentrations (μg m$^{-3}$) between
ARC150X (left)/MID7X (right) and PD simulations.


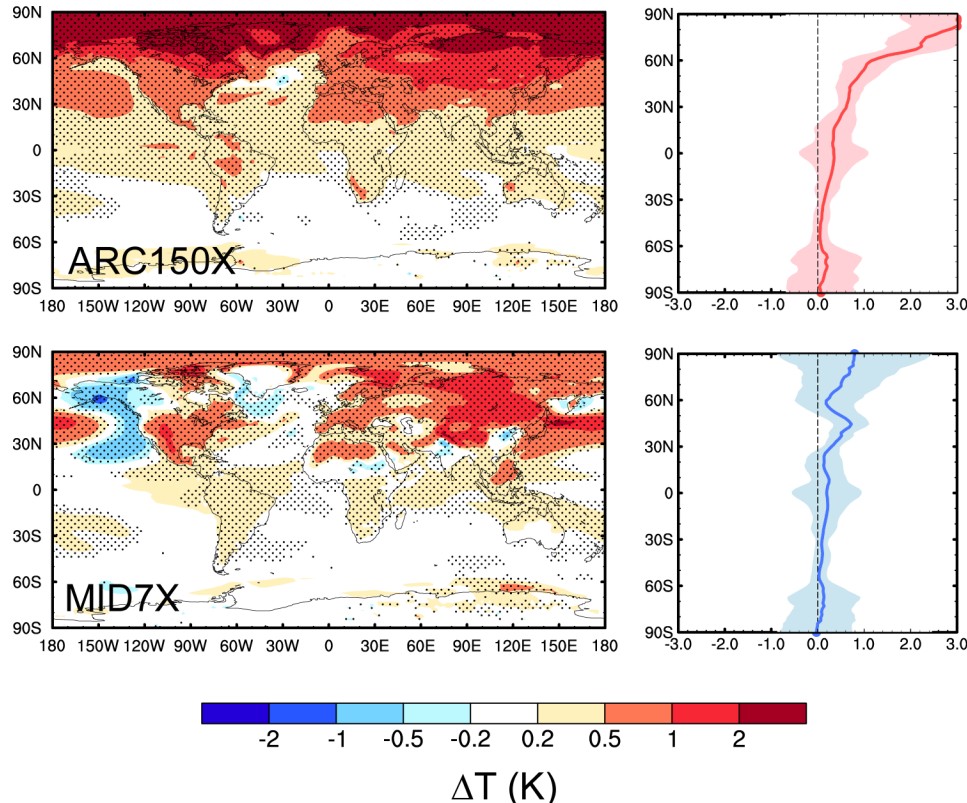

**Figure 2.** Spatial distribution (left) and zonal mean (right) of changes in annual mean
surface air temperature (K) for ARC150X (top) and MID7X (bottom) compared to PD.
The dotted areas in left panels indicate statistical significance with 95% confidence
from a two-tailed Student's t test.

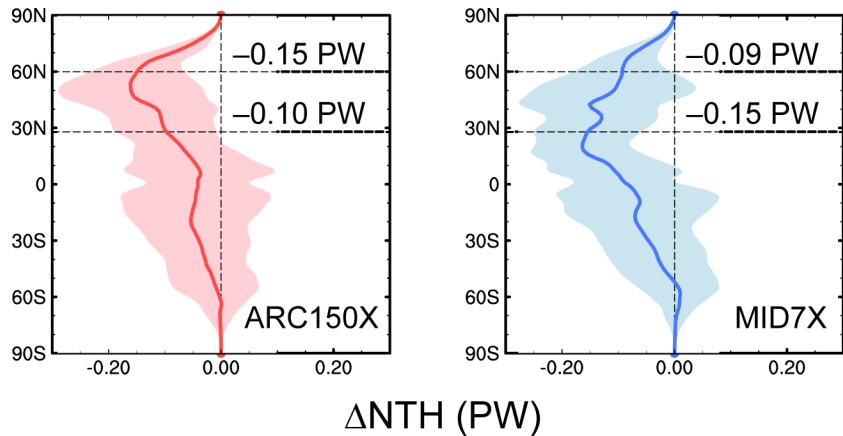



**Figure 3.** Zonal mean of changes in annual mean northward heat transport (NHT,
PW) for ARC150X (left) and MID7X (right) compared to PD. Values of changes in
NHT across 60°N and 28°N are shown in each panel. The shaded areas represent
1-σ for 80-annual means.



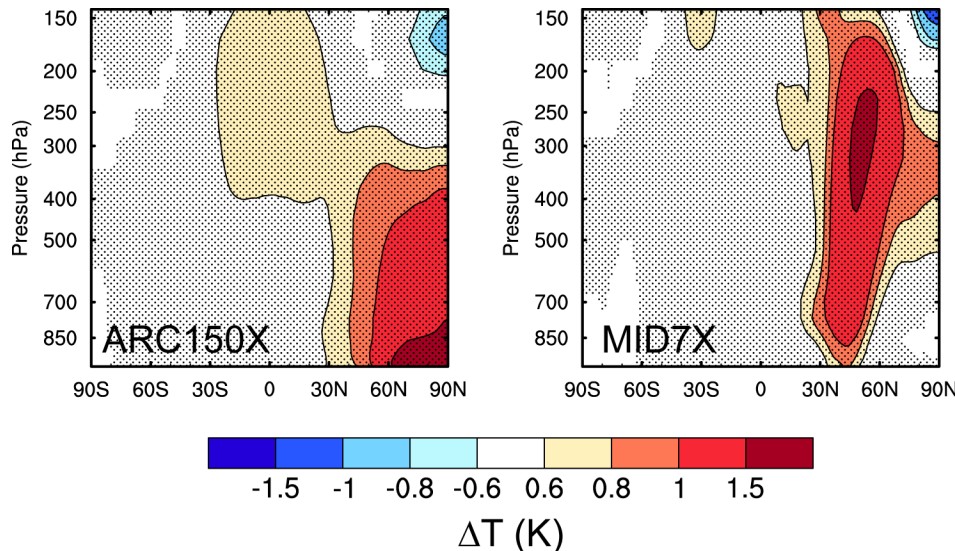

**Figure 4.** Changes in annual and zonal mean temperature (K) for ARC150X (left) and MID7X (right) compared to PD. The dotted areas indicate statistical significance with 95% confidence from a two-tailed Student's t test.

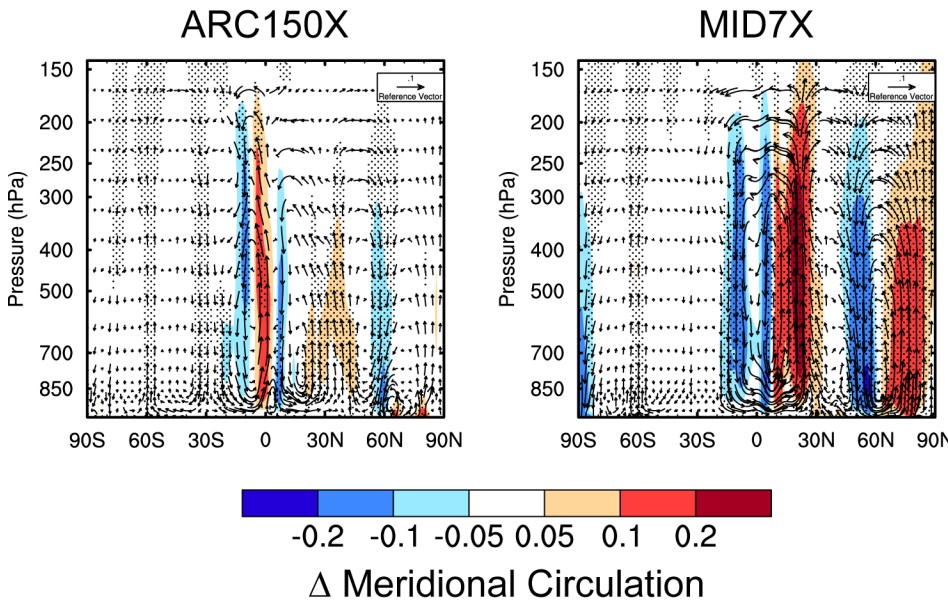

**Figure 5.** Changes in annual and zonal mean meridional wind vectors (m s$^{-1}$) and
vertical velocity (contours; Pa s$^{-1}$ scaled by a factor of −100) for ARC150X (left) and
MID7X (right) compared to PD. The dotted areas indicate statistical significance with
95% confidence from a two-tailed Student's t test.

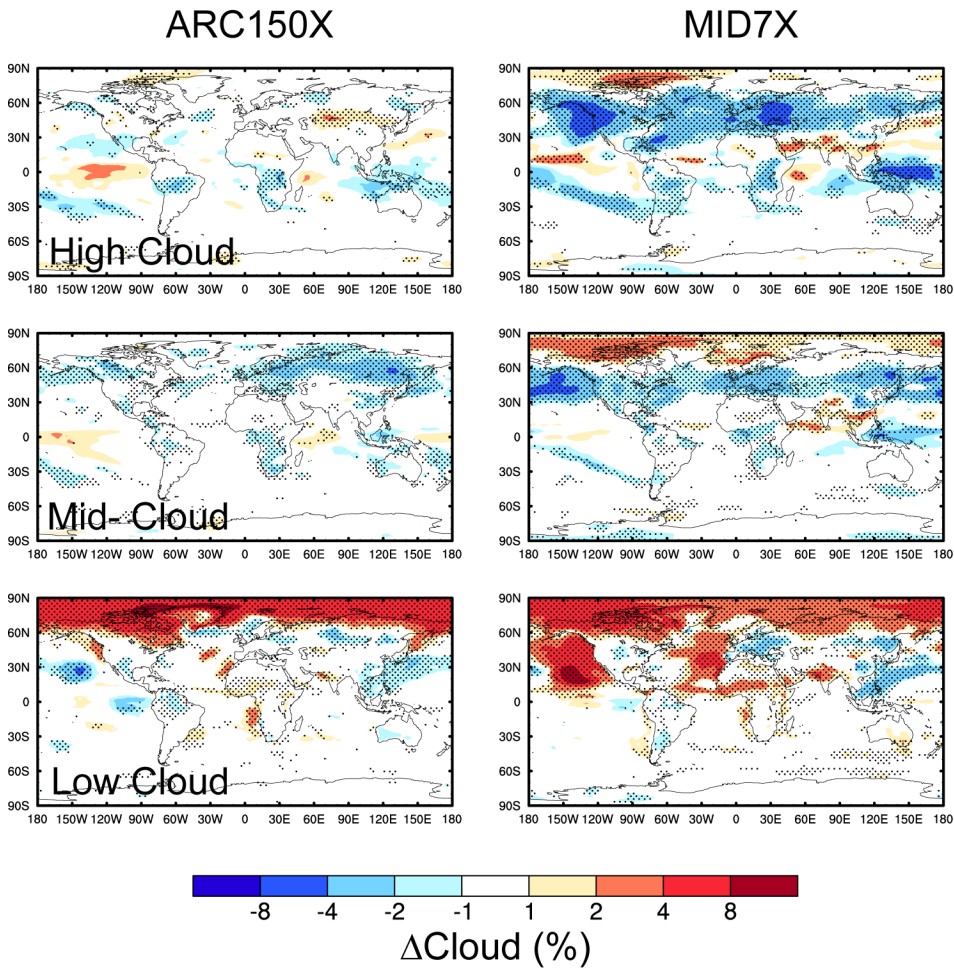

**Figure 6.** Changes in annual mean high (top), mid-level (middle), and low (bottom) cloud fraction (%) for ARC150X (left) and MID7X (right) compared to PD. The dotted areas indicate statistical significance with 95% confidence from a two-tailed Student's t test.

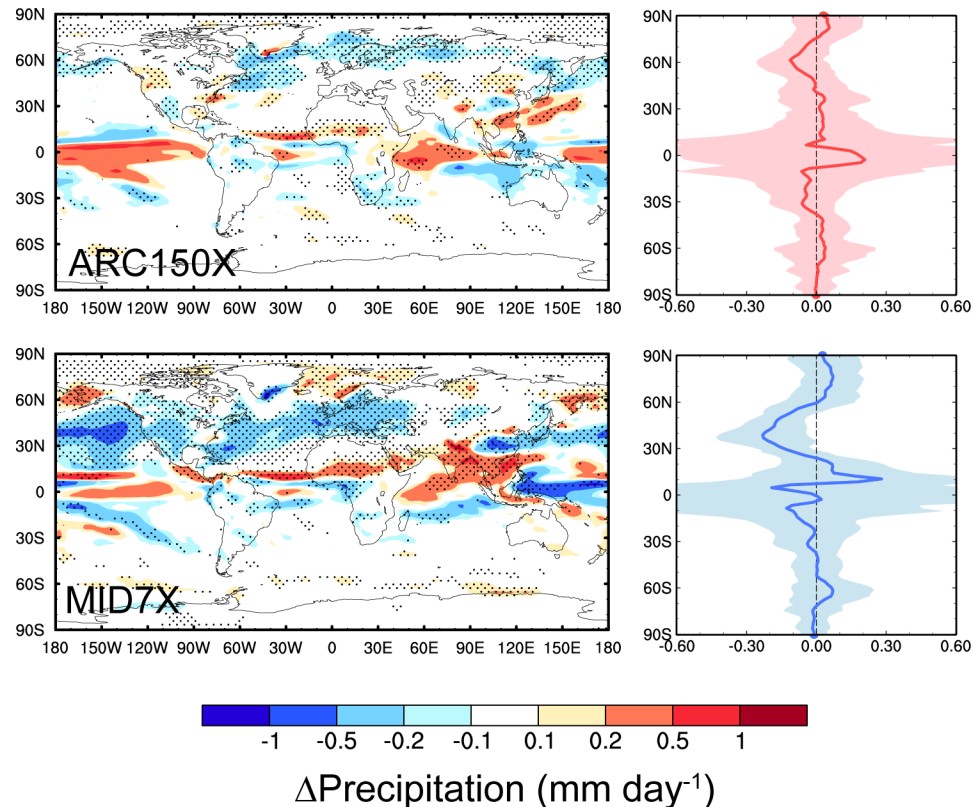

**Figure 7.** Spatial distribution (left) and zonal mean (right) of changes in annual mean total precipitation rate (mm day$^{-1}$) for ARC150X (top) and MID7X (bottom) compared to PD. The dotted areas in left panels indicate statistical significance with 95% confidence from a two-tailed Student's t test.

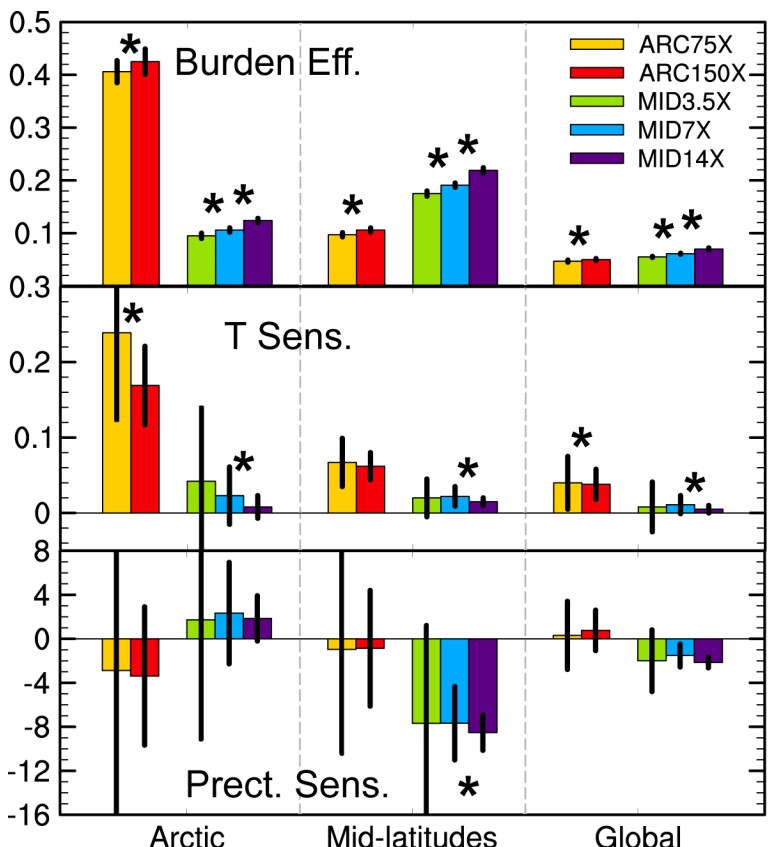

**Figure 8.** Burden efficiencies, temperature and precipitation sensitivities over the Arctic, mid-latitudes and the whole globe for ARC75X, ARC150X, MID3.5X, MID7X and MID14X. Burden efficiencies, temperature sensitivity and precipitation sensitivity are calculated as changes in regional mean BC column burden, surface temperature and total precipitation rate divided by changes in global total BC emissions between perturbed and PD simulations, respectively. Error bars represent 1-σ for 80-annual means. Asterisk between two bars (ARC75X/ARC150X, MID3.5X/MID7X, and MID7X/MID14X) indicates statistically significant changes with 95% confidence from a two-tailed Student's t test.

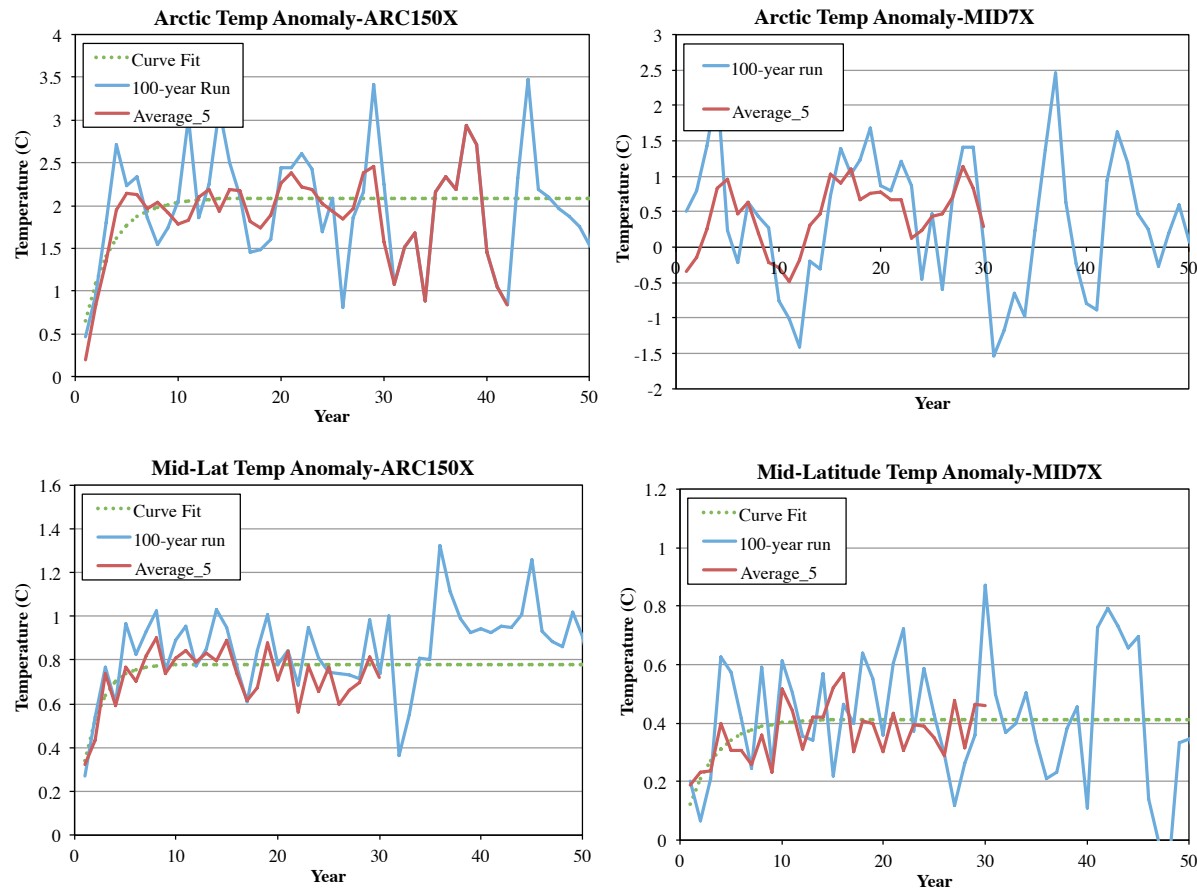

979
980
**Figure 9.** Time series of mean surface temperature response from ARC150X (left) and MID7X (right) BC emission perturbations as compared to PD. The response is shown over the Arctic (top) and mid-latitudes (bottom). Shown are the 100-year ensemble simulation (blue lines), the average of five 30-year ensemble members (red), and a numerical fit for an exponential approach to the long-term average (green dashed line). Curve fits used the package STAN in R, which is Bayesian inference using the No-U-Turn sampler. Note that MID7X Arctic temperature response does not result in a fit due to noise.


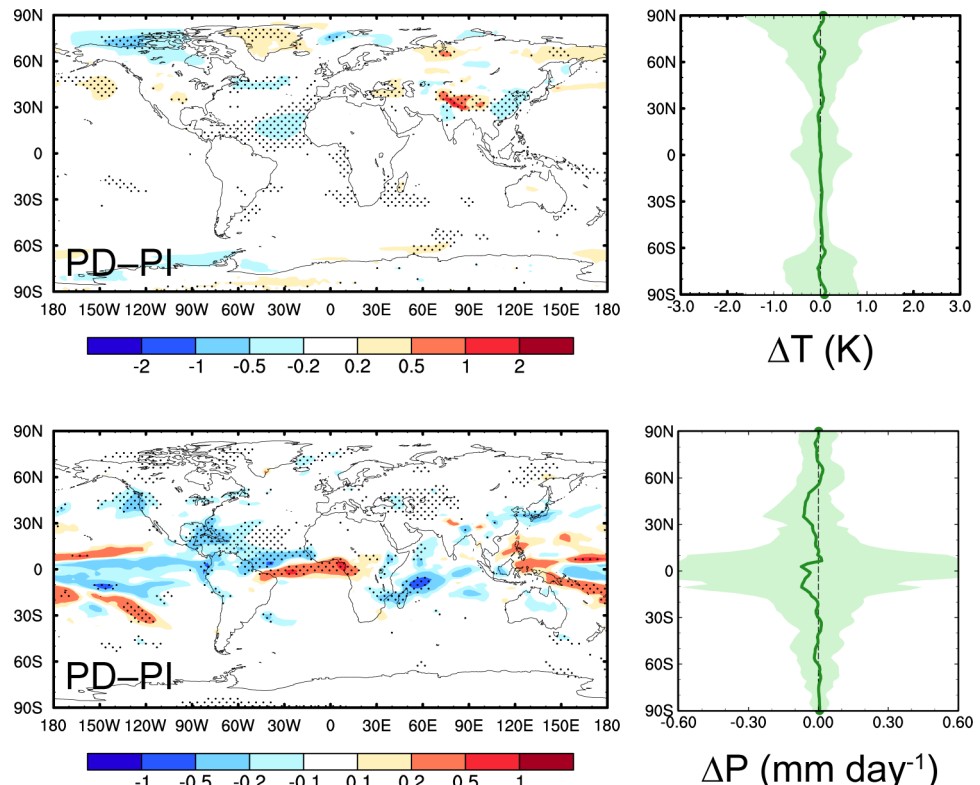

**Figure 10.** Spatial distribution (left) and zonal mean (right) of differences in annual
mean surface temperature (K, top) and total precipitation rate (mm day⁻¹, bottom)
between PD and PI. The dotted areas in left panels indicate statistical significance
with 95% confidence from a two-tailed Student's t test.