# Peer review of "Variability, timescales, and non-linearity in climate responses to black carbon emissions"

_Atmospheric Chemistry and Physics, 2018_

## Referee Comment (RC1) · Anonymous Referee #1 · 10 Oct 2018

This study concerns the climate response of black carbon (BC) emissions in a fully-coupled climate model. The study is motivated by the potential for BC emissions reductions, and the authors evaluate the non-linearity of emission perturbations and the transient response. As BC has received attention from a policy-perspective to reduce global warming (e.g. CCAC and the Arctic Council), investigating possible non-linearities in emission perturbations and the transient response of BC is important and highly relevant for ACP.

Unfortunately, I think the study fails to answer these questions. Since the authors do not find any significant climate change from present-day BC emissions, the authors conclude that BC emission cuts may not be detectable and that the climate impact of BC should be expressed directly in terms of emissions.

[Figure]

First; Do the authors mean that we cannot say anything about the climate impact of BC? I would argue that emissions are not climate impacts. As this paper is clearly motivated by policy-relevant questions, I am confused about what the authors are trying to conclude.

Second; I agree that cutting BC emissions would not be detectable on a thermostat in the real world. As a matter of fact; few things would, except radical changes like cutting CO2-emissions to zero. Is this relevant? As researchers, we must use models (this is why we use them) to provide our best estimate on the climate impact of e.g. cutting BC emissions, and then it is up to the policy-makers to decide if it is worth it in terms of costs, feasibility, co-emitted species etc.

Third; the simulating period for these runs are too short to make these conclusions. If the simulation period was long enough, I argue that the authors would 1) be able to detect a signal from present-day emissions and 2) quantify the non-linearity of different emissions perturbations. Are the temperature sensitivities in Figure 8 significantly different from each other? The authors refer to natural variability as error bars, which I find a bit odd.

The most important finding in this study, I think, is the short transient response of 2-3 years and the lack of a long-term response that the authors find for BC . This contradicts the much-used study by Boucher and Reddy (2007) where it is shown that BC both has a short-term response and a long-term response (ocean). If this is true it will be important for policy-makers, as a rapid BC mitigation will not be crucial for reaching e.g. the 2-degree target and can be delayed for some time. Physically, this means that BC emissions mostly influences the boundary layer over land surface, and do not warm the ocean due to a stabilization of the marine stratocumulus clouds. Would this be specific or sensitive to the model and the cloud scheme? In Boucher and Reddy (2007), they use an impulse temperature response function with both a short-term and a long term. How certain are the authors that there is no long-term response? In L404 you state that 'by our observation that there is no detectable long-term trend after the

initial transient period'. This is a bit vague. Can you perform a hypothesis test to see if there is no long-term trend? But, again, the simulation period is too short to estimate any long-term responses.

I suggest that the authors either extend their simulation period or significantly tone down their conclusions. But if the latter; I am not sure how much added value this study will provide. However, if the authors do extend their simulation period (yes, this will require some extra work), I think this study can be an important contribution to the field.

---

## Referee Comment (RC2) · Anonymous Referee #2 · 13 Nov 2018

This study investigated the regional climate responses, non-linearity, and short-term transient responses to BC emission. The topic is of interest and the method scientifically sounds. I have a few comments: Major comments: 1)There is no model evaluation in this study. How does the model in terms of the aerosol species or climate variables? Some statistical evaluations are useful to warrant the confidence in interpreting the model results. 2) The authors mainly analyzed the results from the annual scale. Are there any substantial differences in a finer temporal scale, i.e., daily or monthly or seasonal? Minor comments: 2)Line 283: due to their 30 times larger: how to get the value of 30? 3) Figures 4-7: The captions need to be revised to explain the meaning of the dots. In Figure 2, it says that "The dotted areas in left panels indicate statistical significance with 95% confidence from a two-tailed Student's t test." Probably the dots

in figures 4-7 share the same meaning. Please clarify. 4) Page 10, Line 194: Both mass and number of BC 5) Line 273-274: from Figure 3, we can see some substantial changes in the south hemisphere. What does this mean? Were the emissions scaled in the south hemisphere as well? 6) Line 446-447: Large scale surface temperature from current-day BC emissions is statistically indistinguishable from zero. The authors' statement is based on global scale. Since the variability is large, are there any features (i.e., larger change in some areas) in different locations? 7) Line 522: BC direct radiative effects and snow/ice-albedo forcings have much larger signal to noise ratios: Could you please explain a bit more what the larger signal to noise ratio mean?
* * *

---

## Author Comment (AC1) · 16 Jan 2019

**Manuscript # acp-2018-904**

**Responses to Reviewer #1**

This study concerns the climate response of black carbon (BC) emissions in a fully- coupled climate model. The study is motivated by the potential for BC emissions reductions, and the authors evaluate the non-linearity of emission perturbations and the transient response. As BC has received attention from a policy-perspective to reduce global warming (e.g. CCAC and the Arctic Council), investigating possible non-linearities in emission perturbations and the transient response of BC is important and highly relevant for ACP.

We thank the reviewer for all the insightful comments. Below, please see our point-by-point responses (in blue) to the specific comments and suggestions and the changes that have been made to the manuscript to take into account all the comments raised here.

Unfortunately, I think the study fails to answer these questions. Since the authors do not find any significant climate change from present-day BC emissions, the authors conclude that BC emission cuts may not be detectable and that the climate impact of BC should be expressed directly in terms of emissions.

Indeed, we do conclude that there may be no detectable surface **temperature** change impacts from present-day BC over much of the world (note we do not conclude there are no *climate* impacts, since BC will impact other variables). Note, our specific wording on this point:

These results indicate that even substantial BC emissions reductions from current levels may lead to detectable surface temperature changes for only limited regions of the globe. (line 560)

Which we believe accurately reflects our findings and does, indeed, address the questions raised by many on this topic (albeit from one model of course).

We did not intend to say that the impact of BC should be expressed as emissions, we apologize for the misunderstanding. This portion of the discussion section has been re-written to clarify as:

We suggest that impacts of BC on climate should be expressed directly in terms of impacts per unit emissions (e.g. table 1), and not only relative to forcing given the complex relationship between BC climatic impacts and top of the atmosphere forcing. In addition, BC impacts should be re-evaluated using coupled models, and provided with measures of response variability, such as standard deviation. First; Do the authors mean that we cannot say anything about the climate impact of BC? I would argue that emissions are not climate impacts. As this paper is clearly motivated by policy-relevant questions, I am confused about what the authors are trying to conclude.

**Response:**

We can indeed say something about BC impacts on climate. We can say that, over most of the globe, BC impacts on surface temperature are very small.

Second; I agree that cutting BC emissions would not be detectable on a thermostat in the real world. As a matter of fact; few things would, except radical changes like cutting CO2-emissions to zero. Is this relevant? As researchers, we must use models (this is why we use them) to provide our best estimate on the climate impact of e.g. cutting BC emissions, and then it is up to the policy-makers to decide if it is worth it in terms of costs, feasibility, co-emitted species etc.

**Response:**

This statement by the reviewer that only changes such as cutting CO2 emissions to zero would be detectable is not correct. For example, differences in climate variables between a RCP8.5 and RCP4.5 scenario begin to be statistically detectable as early as 25–30 years after scenarios diverge (Tebaldi and Friedlingstein 2013), well before CO2 emissions are zero. Cutting SO2 emissions to zero, for example, also leads to detectable changes in surface temperature over much of the world (Baker et al. 2015). However, the impact of reducing BC emissions on surface temperature is much smaller due, in part, to the counteracting cooling and warming effects of fast feedbacks that are only present in coupled model simulations (as demonstrated in a number of models recently by Stjern et al. 2017).

Note that we did not provide any recommendation on the desirability of reducing BC emissions. We have, however, found in our simulations that the impact of reducing BC emissions, even to the point of eliminating all anthropogenic emissions, is small and statistically undetectable across most of the globe.

Third; the simulating period for these runs are too short to make these conclusions. If the simulation period was long enough, I argue that the authors would 1) be able to detect a signal from present-day emissions and 2) quantify the non-linearity of different emissions perturbations. Are the temperature sensitivities in Figure 8 significantly different from each other? The authors refer to natural variability as error bars, which I find a bit odd. Response:

The runs were as long as computationally feasible (e.g. multiple 100-year coupled simulations), and our results are provided with the uncertainty

estimates derived from these simulations. To reduce uncertainty further, a far longer set of runs would be required since noise in this case would reduce as 1/sqrt(Neff), where Neff equals the simulation length divided by the correlation time-scale for the processes in question.

We have added statistical tests in Figure 8 and Table 2 as shown below and modified the manuscript adding these tests.

Note that we did not "refer to natural variability as error bars", the error bars represent one standard deviation of the signals derived from our runs to indicate uncertainty. We do, however, conclude that the root of this variability in our signal likely stems from internal variability in the model due to the statistical properties we find in our analysis.

[revised manuscript text omitted]

The most important finding in this study, I think, is the short transient response of 2-3 years and the lack of a long-term response that the authors find for BC. This contradicts the much-used study by Boucher and Reddy (2007) where it is shown that BC both has a short-term response and a long-term response (ocean). If this is true it will be important for policy-makers, as a rapid BC mitigation will not be crucial for reaching e.g. the 2-degree target and can be delayed for some time. Physically, this means that BC emissions mostly influences the boundary layer over land surface, and do not warm the ocean due to a stabilization of the marine stratocumulus clouds. Would this be specific or sensitive to the model and the cloud scheme? In Boucher and Reddy (2007), they use an impulse temperature response function with both a short-term and a long term. How certain are the authors that there is no long-term response? In L404 you state that 'by our observation that there is no detectable long-term trend after the initial transient period'. This is a bit vague. Can you perform a hypothesis test to see if there is no long-term trend? But, again, the simulation period is too short to estimate any long-term responses. Response:

We note that Boucher and Reddy (2008), and much other previous work, use a GHG impulse response function for the BC response. This was an implicit assumption that the system responds similar to a BC impulse as to a GHG inputs. This is known not to be the case for aerosols in general, since their forcing is not uniformly distributed in space (e.g., the "geometric effect" as noted by Meinshausen et al. (2011) and Shindell (2014)). As we note in the text, a similar result for BC (for a global emission perturbation) was found by Sand et al. (2015).

We did indeed statistically test for a long-term response and we have amended the text to cite the result of our linear fit showing there is no long-term response. We have amended the text to indicate that, as the reviewer points out, that we can only conclude there is no significant response over a 100-year time horizon. We cannot draw conclusions for longer-time scales.

I suggest that the authors either extend their simulation period or significantly tone down their conclusions. But if the latter; I am not sure how much added value this study will provide. However, if the authors do extend their simulation period (yes, this will require some extra work), I think this study can be an important contribution to the field.

**Response:**

We disagree that it is necessary to extend the simulations to draw relevant conclusions. While it would be interesting to do so, it is not clear if the computational costs could be justified. 100-years is a standard length for model diagnoses of this sort (e.g. Baker et al. 2015) and provides sufficient statistics for analysis.

---

## Author Comment (AC2) · 16 Jan 2019

**Manuscript # acp-2018-904**

**Responses to Reviewer #2**

This study investigated the regional climate responses, non-linearity, and short-term transient responses to BC emission. The topic is of interest and the method scientifically sounds.

We thank the reviewer for all the insightful comments. Below, please see our point-by-point responses (in blue) to the specific comments and suggestions and the changes that have been made to the manuscript, attempting to take into account all the comments raised here.

Major comments

1)There is no model evaluation in this study. How does the model in terms of the aerosol species or climate variables? Some statistical evaluations are useful to warrant the confidence in interpreting the model results.
Response:
    Previous studies have extensively evaluated the CAM5 model simulations of concentration, deposition, vertical profile and optical properties of BC (Wang et al., 2013; Wang et al., 2015; Zhang et al., 2015a,b; Liu et al., 2016; Yang et al., 2017, 2018a,b) ), as well as climate variables (Hurrell et al., 2013; Yang et al., 2016a,b). The model can simulate well the BC aerosol and climate variables in most regions of the globe, but was reported to underestimate BC concentrations over China (Yang et al., 2018a) and the Arctic (Wang et al., 2013) (although this earlier study used a different emissions dataset), implying a possible underestimate of climate responses to BC emissions in this study. We have added these texts in the methods section.

2) The authors mainly analyzed the results from the annual scale. Are there any substantial differences in a finer temporal scale, i.e., daily or monthly or seasonal?
Response:
    Thanks for the suggestion. We have added Figures S2 and S3 as below to show the seasonal and monthly surface air temperature change. Since that the climate response and variability are normally longer than daily scale, we skipped the daily information. We have also added texts in the manuscript to illustrate the seasonal difference that "The seasonal mean surface air temperature responses present similar spatial patterns (Figure S2), but slightly different magnitudes (Figure S3). Over the Arctic, the warming due to Arctic BC emissions is weakest in boreal summer. This is because the smaller summer sea ice and snow fraction in the Arctic weakens BC snow/ice-albedo forcing.

However, in the mid-latitudes, warming is strongest in boreal summer for both Arctic and mid-latitude BC emissions, because of stronger summer solar insolation and, therefore, stronger BC heating in the atmosphere."

[Figure]

**Figure S2.** Spatial distribution of changes in December-January-February (DJF), March-April-May (MAM), June-July-August (JJA) and September-October-November (SON) (from top to bottom) mean surface air temperature (K) for ARC150X (left) and MID7X (right) compared to PD. The dotted areas indicate statistical significance with 95% confidence from a two-tailed Student's t test.

[Figure]

**Figure S3.** Changes in Arctic (top), mid-latitude (middle) and global (bottom) monthly mean surface temperature (K) for ARC150X/MID7X compared to PD.

Minor comments:

2)Line 283: due to their 30 times larger: how to get the value of 30?
Response:
    Revised to "Mid-latitude emissions, however, are likely to have a larger present-day impact overall due to their 35 times larger preindustrial to present-day emission increase (2.874 Tg yr$^{-1}$) than Arctic emissions (0.082 Tg yr$^{-1}$)."

3) Figures 4-7: The captions need to be revised to explain the meaning of the dots. In Figure 2, it says that "The dotted areas in left panels indicate statistical significance with 95% confidence from a two-tailed Student's t test." Probably the dots in figures 4-7 share the same meaning. Please clarify.
Response:
    Yes. Clarified in Figure 4-7.

4) Page 10, Line 194: Both mass and number of BC
Response:
    CAM5 predicts both mass and number mixing ratios of aerosols, which requires both the mass and number emissions. That is why we mentioned here that "Both mass and number of BC emissions are perturbed proportionally."

5) Line 273-274: from Figure 3, we can see some substantial changes in the south hemisphere. What does this mean? Were the emissions scaled in the south hemisphere as well?
Response:
    We only scaled BC emissions in the Arctic and mid-latitudes of the Northern Hemisphere, respectively. Energy balance is not like the short-lived BC aerosol, which is mainly located near its sources. With the strong BC perturbation in the whole Arctic and mid-latitudes, energy balance will be changed not only in the regions where it is perturbed, but in the global scale, because the perturbation changes the latitudinal temperature gradient and therefore the poleward heat transport in both Northern and Southern Hemisphere.

6) Line 446-447: Large scale surface temperature from current-day BC emissions is statistically indistinguishable from zero. The authors' statement is based on global scale. Since the variability is large, are there any features (i.e., larger change in some areas) in different locations?
Response:
    Thanks for the suggestions. Yes, there are significant temperature changes regionally. "PD emissions produce statistically significant surface air temperature changes over only limited regions in the Northern Hemisphere. Decreased temperatures are found over eastern China, South Asia, North

Atlantic Ocean, and North American Arctic, partly due to cloud changes driven by BC rapid adjustments. Increased temperatures are found over the Tibetan Plateau, Greenland and high-latitude land regions likely because of the BC snow/ice-albedo effect". To make it clearer, we have also revised the text as "Although statistically significant surface temperature changes are found regionally, as mentioned above, large-scale global surface temperature change from current-day BC emissions is statistically indistinguishable from zero".

7) Line 522: BC direct radiative effects and snow/ice-albedo forcings have much larger signal to noise ratios: Could you please explain a bit more what the larger signal to noise ratio mean?
Response:
    Changed to "aerosol burdens, BC direct radiative effects and snow/ice-albedo forcings have much larger signal to noise ratios, i.e. ratio of mean response to standard deviation (Table 1)"

Reference:

Hurrell, J. W., Holland, M. M., Gent, P. R., Ghan, S., Kay, J. E., Kushner, P. J., Lamarque, J. F., Large, W. G., Lawrence, D., Lindsay, K., Lipscomb, W. H., Long, M. C., Mahowald, N., Marsh, D. R., Neale, R. B., Rasch, P., Vavrus, S., Vertenstein, M., Bader, D., Collins, W. D., Hack, J. J., Kiehl, J., and Marshall, S. (2013), The Community Earth System Model A Framework for Collaborative Research, B. Am. Meteorol. Soc., 94, 1339–1360, doi:10.1175/BAMS-D-12-00121.1.

Liu, X., Easter, R. C., Ghan, S. J., Zaveri, R., Rasch, P., Shi, X., Lamarque, J.-F., Gettelman, A., Morrison, H., Vitt, F., Conley, A., Park, S., Neale, R., Hannay, C., Ekman, A. M. L., Hess, P., Mahowald, N., Collins, W., Iacono, M. J., Bretherton, C. S., Flan- ner, M. G., and Mitchell, D.: Toward a minimal representation of aerosols in climate models: description and evaluation in the Community Atmosphere Model CAM5, Geosci. Model Dev., 5, 709–739, doi:10.5194/gmd-5-709-2012, 2012.

Liu, X., Ma, P.-L., Wang, H., Tilmes, S., Singh, B., Easter, R. C., Ghan, S. J., and Rasch, P. J. (2016), Description and evaluation of a new four-mode version of the Modal Aerosol Module (MAM4) within version 5.3 of the Community Atmosphere Model, Geosci. Model Dev., 9, 505–522, doi:10.5194/gmd-9-505-2016.

Meinshausen, M., Raper, S. C. B., and Wigley, T. M. L.: Emulating coupled atmosphere-ocean and carbon cycle models with a simpler model, MAGICC6 – Part 1: Model description and calibration, Atmos. Chem. Phys., 11, 1417-1456, https://doi.org/10.5194/acp-11-1417-2011, 2011.

Shindell, D.T., 2014: Inhomogeneous forcing and transient climate

sensitivity. *Nature Clim. Change*, **4**, 274-277, doi:10.1038/nclimate2136

Tebaldi, C. and Friedlingstein, P., 2013. Delayed detection of climate mitigation benefits due to climate inertia and variability. *Proceedings of the National Academy of Sciences*, p.201300005.

Wang, H., Easter, R. C., Rasch, P. J., Wang, M., Liu, X., Ghan, S. J., Qian, Y., Yoon, J.-H., Ma, P.-L., and Vinoj, V.: Sensitivity of remote aerosol distributions to representation of cloud–aerosol interactions in a global climate model, Geosci. Model Dev., 6, 765-782, doi:10.5194/gmd-6-765-2013, 2013.

Wang, M., Larson, V., Ghan, S., Ovchinnikov, M., Schanen, D., Xiao, H., Liu, X., Guo, Z., and Rasch, P.: A multiscale modeling framework model (superparameterized CAM5) with a higher-order turbulence closure: Model description and low-cloud simulations, J. Adv. Model. Earth Syst., 7, 484–509, doi:10.1002/2014MS000375, 2015.

Yang, Y., Russell, L. M., Xu, L., Lou, S., Lamjiri, M. A., Somerville, R. C. J., Miller, A. J., Cayan, D. R., DeFlorio, M. J., Ghan, S. J., Liu, Y., Singh, B., Wang, H., Yoon, J.-H., and Rasch, P. J.: Impacts of ENSO events on cloud radiative effects in preindustrial conditions: Changes in cloud fraction and their dependence on interactive aerosol emissions and concentrations, J. Geophys. Res. Atmos., 121, 6321–6335, doi:10.1002/2015JD024503, 2016a.

Yang, Y., Russell, L. M., Lou, S., Lamjiri, M. A., Liu, Y., Singh, B., and Ghan, S. J.: Changes in Sea Salt Emissions Enhance ENSO Variability, J. Climate, 29, 8575–8588, doi:10.1175/JCLI-D-16-0237.1, 2016b.

Yang, Y., Wang, H., Smith, S. J., Ma, P.-L., and Rasch, P. J.: Source attribution of black carbon and its direct radiative forcing in China, Atmos. Chem. Phys., 17, 4319-4336, doi:10.5194/acp-17-4319-2017, 2017.

Yang, Y., Wang, H., Smith, S. J., Zhang, R., Lou, S., Qian, Y., Ma, P.-L., Rasch, P. J.: Recent intensification of winter haze in China linked to foreign emissions and meteorology, Sci. Rep., 8, 2107, doi:10.1038/s41598-018-20437-7, 2018a.

Yang, Y., Wang, H., Smith, S. J., Zhang, R., Lou, S., Yu, H., Li, C., and Rasch, P. J.: Source apportionments of aerosols and their direct radiative forcing and long-term trends over continental United States, Earth's Future, 6, 793–808, doi:10.1029/2018EF000859, 2018b.